# Biological Effects of Human Exposure to Environmental Cadmium

**DOI:** 10.3390/biom13010036

**Published:** 2022-12-24

**Authors:** Massimiliano Peana, Alessio Pelucelli, Christos T. Chasapis, Spyros P. Perlepes, Vlasoula Bekiari, Serenella Medici, Maria Antonietta Zoroddu

**Affiliations:** 1Department of Chemical, Physical, Mathematical and Natural Sciences, University of Sassari, 07100 Sassari, Italy; 2Institute of Chemical Biology, National Hellenic Research Foundation, 11635 Athens, Greece; 3Department of Chemistry, University of Patras, 26500 Patras, Greece; 4School of Agricultural Science, University of Patras, 30200 Messolonghi, Greece

**Keywords:** cadmium, cadmium toxicity, cadmium exposure, chronic exposure, protein targets

## Abstract

Cadmium (Cd) is a toxic metal for the human organism and for all ecosystems. Cd is naturally found at low levels; however, higher amounts of Cd in the environment result from human activities as it spreads into the air and water in the form of micropollutants as a consequence of industrial processes, pollution, waste incineration, and electronic waste recycling. The human body has a limited ability to respond to Cd exposure since the metal does not undergo metabolic degradation into less toxic species and is only poorly excreted. The extremely long biological half-life of Cd essentially makes it a cumulative toxin; chronic exposure causes harmful effects from the metal stored in the organs. The present paper considers exposure and potential health concerns due to environmental cadmium. Exposure to Cd compounds is primarily associated with an elevated risk of lung, kidney, prostate, and pancreatic cancer. Cd has also been linked to cancers of the breast, urinary system, and bladder. The multiple mechanisms of Cd-induced carcinogenesis include oxidative stress with the inhibition of antioxidant enzymes, the promotion of lipid peroxidation, and interference with DNA repair systems. Cd^2+^ can also replace essential metal ions, including redox-active ones. A total of 12 cancer types associated with specific genes coding for the Cd-metalloproteome were identified in this work. In addition, we summarize the proper treatments of Cd poisoning, based on the use of selected Cd detoxifying agents and chelators, and the potential for preventive approaches to counteract its chronic exposure.

## 1. Introduction

Cadmium (Cd) was discovered in 1817 in Germany by Friedrich Stromeyer as an impurity in zinc carbonate (ZnCO_3_). The name derives from the Latin word *cadmia* and the Greek word *καδμεία*, an older name for the common zinc (Zn) ore, calamine. Cd is naturally present in the Earth’s crust, at an average concentration estimated to be between 0.1 and 0.2 parts per million (ppm) [1]. Atmospheric Cd can form as a result of natural activities, such as spontaneous biomass combustion and volcanic eruptions. Despite its natural presence on the planet, no biological function has been found for Cd in higher organisms [2], while its toxicity is well known and has been the subject of numerous studies [3]. Cd continues to occur in the environment, particularly in aerosol form, as a result of human activities, such as the burning of fossil fuels and wastes, or the process of mining metal ores and industrial emissions, representing a current threat to public health [4]. Humans are protected from chronic exposure to low Cd concentrations by the presence of metallothioneins (MTs), a family of ubiquitous small cysteine-rich proteins, the specific function of which is to regulate the metabolism of Zn. MTs play important roles in protection against ion toxicity from several heavy metals, DNA damage, and oxidative stress. Thanks to the presence of many sulfhydryl groups (–SH), MTs are able of complexing, under tolerable exposures, almost all ingested Cd ions [2]. In the kidneys, the resulting Cd^2+^-complexes are, in part, excreted with the urine. The cysteine content of MTs can be up to 30%. These residues can be arranged in motifs (e.g., Cys-x-Cys or Cys-x-x-Cys), and this aspect is fundamental for the formation of metal-binding clusters. The protective effect of MTs against metal toxicity has been largely discussed in a number of studies [5,6]. Apart from Cd, MTs can bind other metal ions, such as mercury, platinum, etc., in order to protect cells and tissues against their toxicity [7]. Mutant flies with increased quantities of MTs in their genes are able to survive on a diet rich in heavy metals, while normal flies would die from the same diet [8]. An increased expression of these proteins can be preventive against heavy metal intoxication: this expression is regulated by the metal regulatory transcription factor 1 (MTF-1).

If the amount of Cd absorbed by the human body exceeds the complexation capacity of MTs, then there will be an accumulation of the metal, mainly in the kidneys (30%) and liver (30%), with the rest dispersed throughout the other organs and having an extremely long half-life of 10–30 years [9,10]. The biological half-life of Cd in the blood has been estimated to be in the range of 75 to 128 days; however, this half-life mainly represents its deposition in the organs, not the body’s clearance rate [10,11]. Cd bioaccumulation in various tissues affects cell functionality, such as proliferation and differentiation, and is responsible for oxidative stress by the generation of reactive oxygen species (ROS), despite its inability to generate free radicals directly, instead forming them via the replacement of redox-active metal ions, such as Fe^2+^ and Cu^2+^, from their metal-binding sites in proteins. The resulting oxidative stress will induce cellular damage and apoptosis [12]. Cd binding into the mitochondria can inhibit both cellular respiration and oxidative phosphorylation, even at low concentrations [13,14]. With these oxidative mechanisms, Cd has been shown to interfere with almost all major DNA repair systems. It is indeed able to impair nucleotide excision repair (NER), base excision repair (BER), and mismatch repair (MMR) [15]. The disturbance of DNA repair processes may explain the co-mutagenic effects in combination with other carcinogenic sources, as DNA repair systems are not only required for the repair of DNA-induced damage but also for the removal of DNA lesions due to endogenous processes and to keep replication errors low [16]. Once absorbed, the thiol groups of cysteines appear to be the critical targets of Cd ions in proteins, enzymes, and endogenous antioxidants such as glutathione (GSH, L-γ-glutamyl-L-cysteinyl-glycine). The inhibition of antioxidant defense enzymes, such as superoxide dismutase (SOD), lactate dehydrogenase (LDH), catalase (CAT), thioredoxin reductase (TrxR), and glutathione peroxidase (GPx) leads to the further dysregulation of the cellular redox state [3,12,17]. The present paper considers exposure and potential health concerns on environmental cadmium, with particular attention to its carcinogenic action. Moreover, we summarize the strategy of environmental Cd remediation and the proper treatments of Cd poisoning, based on the use of selected detoxifying agents and chelators, either alone or in combination.

## 2. The Bioinorganic Chemistry of Cadmium

Cd (electronic configuration (Kr) 4*d*^10^ 5*s*^2^) belongs to group 12 of the periodic table, together with zinc and mercury, but it is chemically more similar to the former than to the latter. It can be found with oxidation states 0, +1, and +2; however, only Cd^2+^ ions are stable under normal conditions, as well as for Zn. Unlike mercury (Hg), which, with simple anions, form compounds having a covalent character, those of Cd have mainly ionic characteristics. Cd^2+^ has an ionic radius of 0.97 Å, very similar to those of Ca^2+^ (0.99 Å) and Na^+^ (1.09 Å), facilitating their replacement in biological matrices such as bones. Cd^2+^ can form coordination complexes with ligands and, particularly, with biological ones such as proteins and nucleic acids. These interactions are modulated by the innate chemical features of Cd^2+^, which is a soft metal ion in the hard/soft acid-base (HSAB) classification [18,19]. The stereochemistry of Cd^2+^ complexes with ligands in biological fluids is useful for understanding the biological behavior and fate of Cd^2+^ ions. According to the HSAB model, Cd^2+^ is a soft acid having a preference for sulfhydryl donors (soft bases), whereas hard carboxylate/carbonyl aminoacidic groups will generally prefer the borderline Zn^2+^, despite the chemical similarities between these ions [19]. The competition between Cd^2+^ and other divalent cations (Zn^2+^, Mn^2+^, Fe^2+^, Ni^2+^, and Cu^2+^) during complex formation with amino acids, peptides, and chelating agents has been reviewed previously [20].

Cd^2+^ can have a wide range of coordination numbers (from 2 to 8), but it mostly shows the coordination numbers 4 or 6, represented, respectively, by regular tetrahedral and octahedral structures with proteins (Figure 1a,b). Depending on the protein to which Cd is bound, clusters can also be generated. MTs are a lightning example of such polynuclear complexes, as cysteine-rich regions of the proteins are able to cluster Cd^2+^ (Figure 1c).

The geometry distribution of Cd^2+^ in the X-ray structures of those proteins deposited in the Research Collaboratory for Structural Bioinformatics Protein Data Bank (RCSB PDB https://www.rcsb.org/ accessed on 22 October 2022) in which Cd^2+^ forms mononuclear complexes is shown in Figure 2 and relative data are reported in Appendix A. From the analysis of the solved crystal structure available in the database, it transpires that the main coordination geometries around Cd^2+^ are represented by octahedral (43.0%), trigonal bipyramidal (26.7%), and tetrahedral (15.9%) forms (derived geometries were included in the count) [24].

Part of the toxicity of Cd^2+^ is due to its ability to mimic other divalent ions, particularly Zn^2+^ and Ca^2+^, which have a similar radius and a similar valence.

A useful tool to analyze the possibilities that a metal ion has to replace another one in its natural metal coordination sites, is based on the use of a representative minimal functional site (MFS) [25]. The MFS describes the local 3D environment around the metal ion and is independent of the larger context of the protein fold in which it is embedded. In summary, MFSs do not depend on the overall macromolecular structure but instead represent a region crucial for the correct physiological function of a selected metalloprotein [25]. It is possible to extract statistical data from the analysis of MFSs (through the 3D X-ray structures of metal–proteins deposited in the Protein Data Bank), by counting how many amino acidic patterns are common to the different ligands in their coordination binding sites. It is then possible, by crossing these data with the protein selectivity for a given metal ion, to understand how probable the substitution is of the original metal ion with another metal ion. In the case of Cd^2+^, on the basis of data from Metalpdb, the database of metal sites in biological macromolecular structures (https://metalpdb.cerm.unifi.it/, accessed on 22 October 2022) [24], the analysis of the possible cadmium substitution for other metal sites yields the results shown in Figure 3 and Appendix A.

Interestingly, the statistical analysis based on a total of 150,149 PDB structures showed that Cd^2+^ has the probability of substituting for other metal ions in 4189 proteins (~2.8% of the total number of PDB structures). In particular, Cd^2+^ has been shown to have a major affinity for the Zn^2+^ binding sites (1245 proteins, 29.7%), Ca^2+^ (590 proteins, 14.08%), and then Mg^2+^ (610 proteins, 14.9%). While it is widely reported that Cd^2+^ can substitute Zn^2+^ at its metal coordination sites as well as Ca^2+^, the replacement of Mg^2+^ by Cd^2+^ is not well known, nor has it been well studied, due to the huge variety of roles that Mg plays in the human body. It has been suggested that cadmium can interfere with the absorption of magnesium in the gastrointestinal tract, affecting its homeostasis. Conversely, several reports suggest that enhanced dietary magnesium intake can mitigate the pathogenic impact of cadmium exposure and its induced alterations in the homeostasis of zinc, copper, and magnesium itself [26,27].

## 3. Uses and Environmental Dispersion of Cadmium

After its first isolation, Cd was used for a variety of processes, firstly as a painting and plating compound (cadmium sulfide, CdS, was used as a yellowish pigment, and cadmium selenide, CdSe, was used as a red one) for about 100 years, mainly because of the particular pigment’s brightness. Cadmium pigments have been used by artists since the 19th century. Because of their resistance to temperatures up to 3000 °C, these Cd-based pigments can be used for coloring hot pipes or glass; examples include red traffic lights and the lit-up stars on the Moscow Kremlin. During the last years CdS and CdSe semiconductor nanomaterials due to their photoluminescent properties have been extensively studied for their use in various environmental monitoring applications, photovoltaic cells as sensitizers, as well as bio-imaging and nanomedicine [28,29]. The majority of Cd has been used (and still is used) in battery technology, in particular, rechargeable nickel-cadmium (Ni–Cd) batteries. The invention of the rechargeable Ni–Cd battery goes back to 1899 and has played a significant role in electrical technology during the twentieth century. In 2002, the European Union put a limit on the use of Cd in electronics, repealing this action in 2016 and establishing a maximum content of Cd of 0.1%, reducing the limit on Cd MCV (maximum concentration value) in homogeneous material to 0.01% [30]. Unfortunately, only a few countries have restrictions regarding the exportation of Cd components, and even fewer have laws about its recycling. In developing countries, this is becoming a huge problem for public health, where electronic waste (e-waste) recycling activities are often conducted without all the safety measures that are needed [31,32]. Cd was a widely used component of plastics, especially PVC, where it works as a stabilizer, but now it has been completely replaced with less harmful metals and alloys, such as barium–zinc alloy. Another source of cadmium for half a century was from its use as a metrology standard from 1907 to 1960. During that period, the angstrom was defined by fixing the wavelength of the distinct, red spectral line of cadmium at 6438.4696 Å. Due to its ability to efficiently capture neutrons, the metal played a remarkable role in the development of the first nuclear reactors; cadmium-coated rods were used to control the nuclear reaction.

Large amounts of Cd are obtained as byproducts of zinc smelting, as the two metals are often naturally associated in minerals. Therefore, many of the environmental contaminations from Cd occur in areas close to the treatment and smelting plants of zinc, but also those of copper and lead. Cd is released into the environment during the disposal and incineration phases of waste containing, in particular, plastic materials, steel plated with cadmium, and nickel-cadmium batteries. Cd exposure in the workplace takes place during mining and work using Cd-containing ores [33].

## 4. Human Exposure to Cadmium

### 4.1. Ingestion

Cd tends to persist and accumulate in the soil and then enters the metabolism of plants. Its accumulation in edible plant parts, including fruits and seeds, leads to Cd’s entry into the food chain [34]. This accumulation increases with the decrease in pH in soil; consequently, acid rain has the effect of increasing Cd concentrations in plants. The foods that mainly contribute to the daily Cd intake in Western countries include cereals and bread (34%), leafy vegetables, in particular spinach, among adults (20%), potatoes (11%), legumes and nuts (7%), stem/root vegetables (6%), and fruits (5%). In terms of Eastern countries, fish and shellfish can be identified as the major Cd sources, in addition to grains and vegetables, which are represented particularly by rice [35]. A particular danger may be represented by rice grains, which account for more than half of the total Cd intake in Eastern regions of the world, at 44% in Japan [36] and 56% in China [37]. The typical dietary Cd intake has been estimated to be about 30–50 µg/day [38], but normal individuals absorb only a small portion of an orally ingested dose (1–10%) [39]. Although the health risk from dietary cadmium exposure in Eastern nations is generally low, it still remains a cause for concern for some subgroups. In fact, being disseminated across the planet, there are areas with very high levels of Cd in the soil. The crop uptake of Cd in these areas can lead to significant dietary exposures for the people living nearby. For example, in the Jinzu and Kakehashi river basins in Japan, there are areas with soil that is heavily contaminated by Cd, derived from industrial waste [40,41]. Local people who frequently consumed rice cooked with Cd-contaminated water, developed a severe kidney and bone syndrome called “Itai-Itai” disease, characterized by bone deformation and multiple fractures, especially in women [42]. The radiographs showed the presence of osteomalacia and bone decalcification, as well as osteoporosis, and a series of bone deformities due to the replacement of Ca^2+^ by Cd^2+^, with a consequent alteration of the normal bone structure.

It is important to note that in the US, in the late 1960s, the average Cd consumption was estimated to be around 26 μg/day/person [43], and, in the first years of the 1990s, was calculated to be around 18.9 μg/day/person [44]; now, the human Cd intake has been lowered to 4.63 μg/day/person, on average. This corresponds to a value of 75% under the tolerable limit of toxicity. The diminishing of Cd intake in the last 50 years could be attributed to the lowered activities of leaking sewage sludge into agricultural soil, because of more efficient control and environmental awareness in developed countries. These activities were mainly responsible for the transfer of Cd, adsorbed by plants, into the food chain, contributing to increasing human exposure to the metal. Some aquatic organisms can also be largely affected by Cd accumulation reaching levels above regulatory standards. While muscle tissue in fish does not represent a site of accumulation of Cd [45], in commercial oysters (*Crassostrea gigas*) from southern Korea, the mean Cd concentration of 80 samples was found to be 0.591 μg/kg, which was much higher than the mean Cd concentration in water samples, 0.0021 μg/L [46]. The acidification of seawater could increase the accumulation of Cd by most common seafood species, such as *Mytilus edulis*, *Tegillarca granosa*, and *Meretrix meretrix*, with a potential threat to seafood safety [47,48]. Several protein transporters have been identified as having a role in cadmium uptake in the intestinal tract. These include divalent metal transporter 1 (DMT1), cationic amino acid transporter 1 (CaT1), zinc transporters (ZIP4, ZIP8, ZIP14, and ZnT1), copper-transporting P-type ATPase or Menkes ATPase (ATP7A), calcium channel TRVP6, and metallothionein MT-1 and MT-2, for which the increased Cd oral concentrations resulted in relatively increased gene expression [5]. Once introduced into the blood, Cd reaches various organs, such as the kidneys and skeleton, through systemic distribution. Initially, Cd in plasma is bound with high molecular-mass proteins such as albumin, then it is found bound with proteins of the molecular size of MTs, which are believed to be responsible for transporting Cd to the kidneys [5].

### 4.2. Inhalation

Cd air levels can be hundreds of times greater in the workplace than in the general environment. For example, the Occupational Safety and Health Administration (OSHA) fixed the permissible exposure limit (PEL) of Cd fume or Cd oxide in the workplace at 0.1 mg/m^3^, whereas concentrations of Cd in ambient air are 1 × 10^−6^ mg/m^3^ in non-industrialized areas and 4 × 10^−5^ mg/m^3^ in urban areas, respectively. Non-occupational Cd exposure from the air is not expected to pose the risk of adverse health effects. In general, Cd air levels are usually not sufficient to cause health problems among the general population. Even in those areas with high industrial emissions of Cd, its average atmospheric concentration is not higher than 35 ng Cd/m^3^ of air. A Brazilian study showed higher blood Cd levels (~0.22 μg/L) in automotive battery-manufacturing workers than in the control group, who had a mean blood Cd level of around 0.03 μg/L. However, the mean blood Cd level in battery workers was even lower compared to the WHO standard (10 μg/L) [49]. In other words, elevated occupational exposure to Cd decreased in the last 50 years, and such a change is due to better regulations regarding Cd exposure in the workplace. Therefore, the daily deposition rate would be 0.175 ng Cd, assuming that 20 m^3^ of air would be inhaled per day, with a deposition rate of 25% [50]. However, Cd exposure is still considered a threat in workplaces in developing countries, where safety standards are not regulated or respected. The major routes of Cd exposure occurring in the general population are from the ingestion of Cd-contaminated foods and cigarette smoking, since the tobacco plants take up Cd from polluted soil, due to its similarity to Zn [51]. Compared to non-smokers, Cd levels are four to five times higher in the blood and two to three times higher in the kidneys for tobacco smokers [52,53]. Tobacco plants accumulate every polluting metal that might be present in the neighborhood, and, in particular, arsenic (As) since some of its compounds are widely used in the production of pesticides or herbicides [54]. Since both As and Cd are classified by the IARC as belonging to Group 1 of the carcinogenic substances, Cd is believed to be linked to lung cancer in smokers, in synergism with As, as well as with other potentially toxic substances that are eventually present in tobacco leaves.

### 4.3. Permeation

There are negligible amounts of Cd absorbed through the skin; thus, it is not considered to be a critical route of exposure. However, recent research highlighted the environmental significance of photosensitive CdS and CdSe pigments and nano semiconductors, whose oxidized products (cadmium sulfate, CdSO_4_, and cadmium selenite, CdSeO_4_) are considerably more soluble and bioavailable and are, consequently, potentially more dangerous [55]. An in vitro study using human full-thickness skin as a model to characterize the impact of Cd exposure on skin showed that the metal penetrates only the epidermis; it was shown before that its solubility into the stratum corneum layer is a rate-limiting process [56]. Permeability is affected by the Cd concentration applied to the skin. It may also be influenced by pH and metal speciation [19]. The high concentrations of Cd in the epidermis may explain the induction of MTs, as Cd is very effective in activating their expression [56].

## 5. Effect of Human Exposure to Cadmium

### 5.1. General Effects That Are Harmful to Health

In this section, we will describe the main health adverse effects of cadmium exposure, while in the next section, the strictly carcinogenic aspects will be explored. Figure 4 illustrates the main outcomes in health effects following chronic cadmium exposure.

The organs most affected by Cd toxicity are the kidneys; as much as about 30% of body Cd is deposited in the kidney tubule region, provoking tubular damage proportionally to the quantity of Cd not bound to MTs. In one study, diabetics were more susceptible to renal tubular damage from Cd exposure than were the controls [57]. Another common disease that can be correlated to Cd exposure is osteoporosis and/or osteomalacia. Cd has a deleterious impact on bones due to its impairment of Vitamin D uptake in the kidneys. Cd also prevents the absorption of calcium at the gut level, causing general bone disease, such as the aforementioned Itai-Itai disease [58]. Cd has a remarkable role in the reproductive system, where the metal interferes in a number of ways [59]. It has been shown that Cd can alter cell adhesion in the testis, interfering with the normal migration of germ cells across the seminiferous epithelium. Furthermore, decreased testicular growth rate, plasma testosterone, and reduced sperm count and motility have been linked to Cd-induced oxidative stress, as the result of a decrease in GPx, CAT, mitochondrial Mn–SOD, and cytosolic CuZn–SOD [60].

Another organism district that is affected by Cd toxicity is the cardiovascular system. Cd can cause hypertension by the inhibition of the endothelial nitric oxide synthase and the suppression of acetylcholine-induced vascular relaxation [61,62]. Cd could also affect glucose metabolism by acting on a variety of different organs, such as the pancreas, liver, and adrenal gland. Studies suggest a direct effect of Cd on the pancreas, and there is evidence that Cd can alter insulin release from pancreatic β-cells while increasing the activity of all the four enzymes responsible for gluconeogenesis [63]. As has been noticed in Itai-Itai disease, Cd also seems to induce anemia, due to the suppression of erythropoietin production [64]; this mechanism, linked to the suppression of iron transport in the duodenum, may cause iron-deficient anemia [65]. Lungs are also a target organ for Cd toxicity. In chronic Cd exposure, progressive pulmonary fibrosis and impaired lung function with obstructive lung disease may occur [66,67]. Cd also induces neurological dysfunction and brain toxicity, with a complex mechanism [68,69].

Cd could be considered a weak genotoxic and mutagenic agent, due to its low affinity with DNA [70]. A number of toxicogenomic studies confirm the involvement of Cd in the mutation of the following genes: immediate early response genes (IEGs), stress response genes, transcriptional factors, and translational factors [12]. One of the mechanisms by which Cd influences gene expression involves not only the homeostasis of Ca, an element that can be directly mimicked by Cd, but also an element whose concentration inside the cell can be influenced by it [71].

### 5.2. Carcinogenicity of Cadmium

One of the major Cd-induced mechanisms of carcinogenesis is oxidative stress [12]; this is, in part, caused, as mentioned above, by changes in Ca concentrations inside the cell. Cd cannot directly produce ROS because it does not undergo Fenton-like reactions, but, on the other hand, it can replace redox-active metal ions and can inhibit the activity of antioxidant enzymes, as well as promote lipid peroxidation [72]. The major enzymatic antioxidant is SOD, which degrades O_2_∙^−^, and the CAT and GSH redox system, which inactivates H_2_O_2_ and hydroperoxides. Three forms of SOD may be important: Mn-SOD (which is located in the mitochondria), Cu–Zn SOD (which resides in the cytoplasm), and extracellular SOD (which lines the blood vessels). GSH, present in high concentrations in every cell, is able to detoxify Cd in the human blood cell, and its synthesis (triggered by the transcription factor Nrf2) is enhanced after Cd exposure [73]. Cd has been shown to impair global genome nucleotide excision repair (GG-NER): it can interfere with the removal of DNA lesions in cultured mammalian cells caused by benzo[*a*]pyrene and UVC [74]. The underlying mechanism in this impairment has been identified as an interaction with the Zn-binding proteins, which show a common motif (Zn finger), where Zn is complexed to four cysteine and/or histidine residues [16,75,76]. Cd^2+^ can substitute Zn^2+^ in these metal binding sites, inactivating the proteins. Another process of the DNA repair pathway is base excision repair (BER), but it is different from NER. BER, in fact, is activated by a specific class of DNA repair enzymes called glycosylases. Low concentrations of Cd disturb the repair of oxidative DNA base damage, as well as DNA alkylation damage, in mammalian cells [77]. It is important to cite another mechanism that is relevant for maintaining genomic stability; the mismatch repair (MMR) is responsible for the repair of mismatched bases after DNA replication. Cells deficient in MMR usually tend to mutate often and are associated with a greater risk of developing several types of cancer. MMR also has a role in apoptosis, and it has been shown that MMR-deficient cells are about 100 times more resistant to the cytotoxicity of alkylating agents [78]. In extracts of human cells, Cd inhibited at least one step of the MMR process [79]. The underlying mechanism in this inhibition seems to be the interaction of Cd in the process of ATP binding the MMR enzymes, plus their hydrolysis, reducing their binding activity to DNA bases and interfering with their ability to recognize mismatched DNA base pairings [80,81].

Cd intoxication can also lead to cell death by apoptosis with different pathways [82,83]. The extrinsic pathway is initiated by binding the cytokine ligands, such as the Fas ligand (FasL) and the tumor necrosis factor alpha (TNF-ɑ), along with the death receptors, CD95/APO-1 (Fas), and the TNF receptors. Cd can alter the CD95/APO-1 (Fas)/Fas ligand (FasL) signaling pathway, especially in neuronal cells. Moreover, it can increase inflammation markers, including TNF-ɑ and NF-kB, in nephrotic cells, leading to apoptosis by caspase-3 activation. On the other hand, Cd can also influence the intrinsic pathway of apoptosis, increasing the expression of the pro-apoptotic protein, Bax, and suppressing the anti-apoptotic protein, Bcl-2 [84]. In this case, a decrease in the Bcl-2/Bax ratio causes the release of cytochrome c from mitochondria and Ca from the endoplasmic reticulum (ER): this leads to the caspase cascade, with the final activation of caspase-3 and cell death. The activation of caspase-3 has a crucial role in cell death by Cd-mediated apoptosis, so it is logical to hypothesize that the inhibition of caspase-3 can prevent Cd toxicity. There are several studies in this field, claiming that certain natural substances such as taxifolin can protect from Cd apoptosis in skin cells by changing the activity of caspase-3 and -7, and other apoptotic factors [85]. Cd toxicity seems to also play a role in autophagy processes, even if this role has not been understood properly [86]. Cd shows, in fact, a different output for the cell fate in different cell lines. Cd-induced autophagy seems to promote apoptosis in skin epidermal cells [87], in mouse spleen, in neuronal cells [88], in mouse renal tubular epithelial cells, and in rat proximal tubular epithelial cells [89]. On the other hand, Cd-exposed human bronchial epithelial cells (BEAS-2BR) appeared to be autophagy-deficient, down-streaming the anti-apoptotic proteins Bcl-2 and Bcl-xl, apoptosis resistance, and possible carcinogenesis [90].

Cd alters the epigenetic signatures in the DNA of the placenta and of newborns, and some studies indicated marked sexual differences for Cd-related DNA methylation changes. Associations between early Cd exposure and DNA methylation might reflect interference with *de novo* DNA methyltransferases. In particular, the association of cadmium with the DNA methylation of certain CpG sites within the genes of interest in organ development, glucocorticoid synthesis, and cell death has been reported [91,92].

### 5.3. Cd-Metalloproteins with Relevance to Carcinogenesis

According to a recent study, the Cd-metalloproteome might constitute up to 18.4% of the entire human proteome [93]. In the present study, we identified which genes encoding Cd-metalloproteome are genetic markers for 12 cancer types (urothelial bladder carcinoma, breast cancer, colorectal adenocarcinoma, glioblastoma, head and neck squamous cell carcinoma, kidney cancer, acute myeloid leukemia, lung adenocarcinoma, lung squamous cell carcinoma, ovarian cancer, uterine corpus endometrial carcinoma, and multiple myeloma) using the Cancer Genome Atlas (TCGA) pan-cancer repository [94]. Specifically, 12 Cd-binding proteins have been characterized as cancer-related, and their names, abundance, and tissue-specific expression in the human body are listed in Table 1. Figure 5 shows the 12 cancer types associated with each specific gene-encoding Cd-metalloproteome.

Five of these proteins (tyrosine-protein kinase Blk, polypeptide N-acetylgalactosaminyltransferase 10, tyrosine-protein kinase Lck, cAMP-dependent protein kinase catalytic subunit alpha, and TGF-beta receptor type-2) are transferases. B-lymphoid tyrosine kinase (Blk) is an oncogene and a potential target for therapy with dasatinib in cutaneous T-cell lymphoma (CTCL) [95]. Gene knockdown experiments showed that Blk promoted the proliferation of malignant T-cells from CTCL patients, suggesting that Blk may function as an oncogene. Blk is also implicated in childhood acute lymphoblastic leukemia [96].

N-acetylgalactosaminyltransferase genes (GALNTs) and proteins (GalNAcTs) are involved in cancer biology. Aberrant O-glycosylation by GalNAcTs activates a wide range of proteins that carry out the interactions of sessile and motile cells affecting organogenesis and the responses to agonists, stimulating the hyperproliferation and metastization of neoplastic cells [97].

cAMP-dependent protein kinase catalytic subunit alpha is a multi-unit protein kinase that mediates the signal transduction of G-protein-coupled receptors through its activation upon cAMP binding. The cAMP/PKA signaling pathway is altered in different cancers and may be exploited for cancer therapy and/or diagnosis via cell cycle regulation and stimulated cell growth [98].

TGF-beta receptor type-2 TGF (TGFBR2) is the ligand-binding receptor for all members of the TGF-β family. TGF-beta receptor type-2 expression in cancer-associated fibroblasts regulates breast cancer cell growth and survival and is a prognostic marker in pre-menopausal breast cancer [99]. The mutational inactivation of TGFBR2 is the most common genetic event affecting the TGF-β signaling pathway and occurs in ∼20–30% of all colon cancers [100].

Among the above-mentioned 12 Cd-binding proteins, the most abundant, based on the protein abundance database PaxDB (https://pax-db.org/ (accessed on 21 October 2022)) [101], is heat shock protein HSP 90-beta, which is ranked in the top 5% of human proteins (Appendix A). Heat-shock proteins are found at increased levels in many solid tumors and hematological malignancies. Their expression may, in part, account for the ability of malignant cells to maintain protein homeostasis, even in the hostile hypoxic and acidotic microenvironment of the tumor. HSP 90-beta is known as a cancer chaperone, required for the stability and function of multiple mutated, chimeric, and/or over-expressed signaling proteins that promote cancer cell growth and/or survival. It has also been implicated in many other crucial steps of carcinogenesis: the inhibition of programmed cell death and replicative senescence, the induction of tumor angiogenesis, and the activation of invasion and metastasis [102,103,104,105].

A variety of Hsp90 inhibitors have shown antitumor effects as a single agent and in combination with chemotherapy. Current Hsp90 inhibitors are categorized into several classes, based on the distinct modes of inhibition, including (i) the blockade of ATP binding, (ii) the disruption of cochaperone/Hsp90 interactions, (iii) the antagonism of client/Hsp90 associations, and (iv) interference with the post-translational modifications of Hsp90 [106].

## 6. Environmental Remediation of Cadmium

The very first prevention step regarding Cd pollution (which is actually one of the main causes of toxicity for humans) is the removal of Cd from contaminated soil and water. This task can be achieved using both physical and chemical methods: on the one hand, there are physical methods, such as adsorption, ion exchange, and reverse osmosis, while, on the other hand, there are chemical methods, which involve precipitation, electrolysis, and solvent extraction [107]. Besides these classical methods, natural remediation from plants and other organisms has also been investigated. As an environmental pollutant, Cd, as with some other metals, is different from organic pollutants, which can be degraded by microorganisms. Microbial remediation, based on metal biosorption, is currently considered to be an efficient strategy for the detoxification of Cd from contaminated waste, water, soil, and sediments [108,109]. In the past few years, the literature has revealed an outstanding variety of microorganisms that are able to process and eliminate Cd from the environment, such as bacteria, fungi, yeasts, and different kinds of algae. In particular, bacteria such as the genera *Aeromonas, Bacillus*, and *Pseudomonas*, and diatom microalgae are able to efficiently accumulate Cd [110,111].

All these methods can be effective in a controlled environment, but they all need very specific conditions of climate, temperature, and humidity to be carried out in a specific ecosystem [107]. More recently, a different remediation method called “phytoremediation” has been proposed [107,112]. In the literature, it is possible to find a huge number of plant species with the peculiar skill of Cd accumulation; these plants are called Cd hyperaccumulators [113].

## 7. Detoxifying Agents and Chelating Agents for the Prevention and Treatment of Cadmium Toxicity

The toxicity of Cd has also been related to its neurotoxic effects, which develop via the changes that Cd induces in the brain enzymes. Trace elements such as Zn and selenium (Se) have been investigated as preventive agents against Cd toxicity, because of their functional role in the brain. As discussed above, Zn^2+^ possesses chemical and physical characteristics that are similar to Cd^2+^ and naturally compete with it for the binding sites in enzymes. Zn also induces the synthesis of the CNS-specific MT III [114], which has a high affinity for Cd and can cause detoxification by binding it. Selenium is recognized to be effective in improving antioxidant defense, immune functions, and metabolic homeostasis, with a critical role in anti-aging [115]. It plays a role in the depletion of Cd from the body by protecting it against Cd toxicity in a number of different organs, including the brain [116]. Se species combine with Cd ions, and both are excreted out of the body via the bile system. Therefore, there is less Se to form GSH peroxidase, one of the body’s main antioxidants [12]. This results in the formation of greater levels of ROS and hydrogen peroxide, with relative cellular damage. Se supplementation will, therefore, be useful for increasing immunity and effectively restoring the body’s antioxidant defense [115]. Se is also a cofactor of GPx, an antioxidant enzyme that can contribute to reducing Cd oxidative effects [117]. Consequently, a preventive approach to counteract chronic Cd exposure is based on the use of trace element supplements as detoxifying agents. Zn and Mg supplementation also has the potential to modulate and mitigate Cd intoxication in several organs [26,27]. In addition, several studies in animal models showed that vitamins A, C, E are able to decrease the toxic effects of Cd in the kidney, liver, spleen, blood, bones, and brain. However, further human studies are needed to clarify the role of these antioxidant vitamins in reducing Cd-induced toxicity [118].

Another strategy of prevention is action at the adsorption level, in the gastrointestinal tract. Research has indeed demonstrated that some strains of lactic acid bacteria (LAB) may be able to bind and remove heavy metals, such as Cd and lead (Pb) [119]. Moreover, probiotics such as these can have antioxidant properties for the human body, being effective against Cd-induced oxidative stress [120]. One of the most studied and convincing probiotics in this field is the *Lactobacillus plantarum* CCFM8610, which was demonstrated to have good Cd-binding ability and to be capable of protecting the liver and kidneys of mice in acute Cd intoxication [121]. This particular effect is due to the Cd sequestration at the intestinal level, reducing the bioavailability of the metal ion in tissues, and decreasing Cd-induced oxidative stress [122]. It has been demonstrated that *L. plantarum* also has a direct effect of protection against Cd-induced oxidative stress, increasing the MT protein levels in the liver [123]. The role of N-acetyl-cysteine (NAC) has also been investigated in the past few years as a cysteine group donor with a possible key role in the biosynthesis of MTs. It has been discovered that exogenous NAC can increase the production of MTs and enhance the possibility of binding and eliminating Cd^2+^ [124]. Cadmium intoxication from poisoning is extremely uncommon. The literature reported a case of Cd intoxication solved with the concomitant administration of GSH, along with the chelating agent, Ca-EDTA [125]. However, EDTA could increase the Cd levels in the kidneys, leading to renal dysfunction [126], as is the case when using dimercaprol (BAL) [127]. Penicillamine (DPA) has been found not to be efficient in Cd overdoses [128]. Meso-2,3-dimercaptosuccinic acid (succimer, DMSA) is a metal chelator not able to reach the intracellular Cd that is bound to MTs and is stored in the liver and kidneys [129]. However, a water-soluble, lipophilic chelating agent, MiADMSA, a C5-branched alkyl monoester of DMSA, can reach the stored Cd intracellularly, as well as other DMSA derivatives such as Monomethyl DMSA (MmDMSA) and Monocyclohexyl DMSA (MchDMSA) [130]. It has also been reported that a proper combination of selected chelating agents can be considered as more effective than mono-therapy [131]. This strategy considers the concomitant use of DMSA and MiADMSA, or calcium trisodium diethylene triamine pentaacetate (CaDTPA), or NAC [131].

## 8. Conclusions

In summary, this review describes the research associated with human exposure to cadmium, with an emphasis on its biological targets, toxic effects (mainly carcinogenic), and therapeutic approaches. The toxicity of Cd^2+^ is, in part, due to its ability to mimic Zn^2+^ and Ca^2+^ in their biological roles. The presence of metallothioneins protects humans from chronic exposure to low concentrations of Cd^2+^. The mechanisms of Cd-induced carcinogenesis have been related to oxidative stress with the inhibition of antioxidant enzymes, the promotion of lipid peroxidation, and interference with DNA repair systems. Strategies of prevention and treatment of Cd toxicity include the administering of trace elements, e.g., Se, as detoxifying agents, therapeutic schemes involving the use of antioxidant vitamins (A, C, E), and action at adsorption level in the gastrointestinal tract, by means of several probiotics.

Future research directions concerning the topic of this review are predicted to be: (i) an investigation of the important role of other metal-binding proteins, except the metallothioneins, and/or new receptors involved in the binding and transport of Cd^2+^; (ii) advanced molecular profiling of the events associated with Cd carcinogenesis in model systems, which may allow the development of expression signatures for Cd-induced cancers; (iii) elucidation of the mechanisms for Cd^2+^ clustering in the cysteine-rich regions of metallothioneins; (iv) clarification of the mechanisms that are responsible for “phytoremediation”; (v) further studies on the role played by Se in the depletion of Cd from the body; and (vi) development of selective chelating agents for Cd^2+^ removal from aqueous environments via solvent extraction, an area in which our research groups are currently involved [132,133].

## Figures and Tables

**Figure 1 biomolecules-13-00036-f001:**
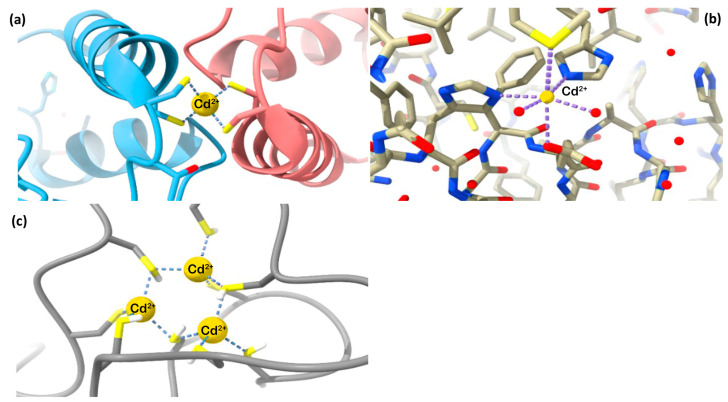
Examples of Cd^2+^ coordination environment in biological systems: (**a**) regular tetrahedral Cd^2+^ complexed with Hah1 metallochaperone protein (Protein Data Bank PDB 1FE0, [21]); (**b**) regular octahedral Cd^2+^ complexed with cytochrome c oxidase (PDB 2EIK, [22]); (**c**) a cluster of Cd^2+^ complexed with metallothionein-1 (PDB 1DFT, [23]).

**Figure 2 biomolecules-13-00036-f002:**
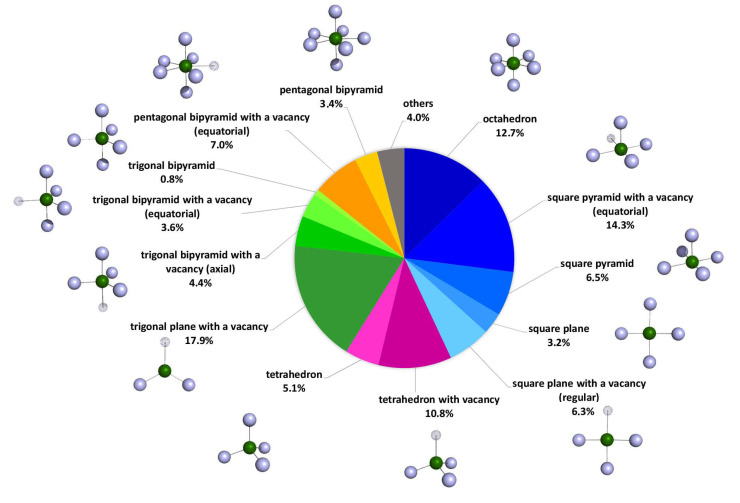
Percentage distribution of geometries for mononuclear Cd^2+^ complexes in proteins, according to the Research Collaboratory for Structural Bioinformatics Protein Data Bank (RCSB PDB).

**Figure 3 biomolecules-13-00036-f003:**
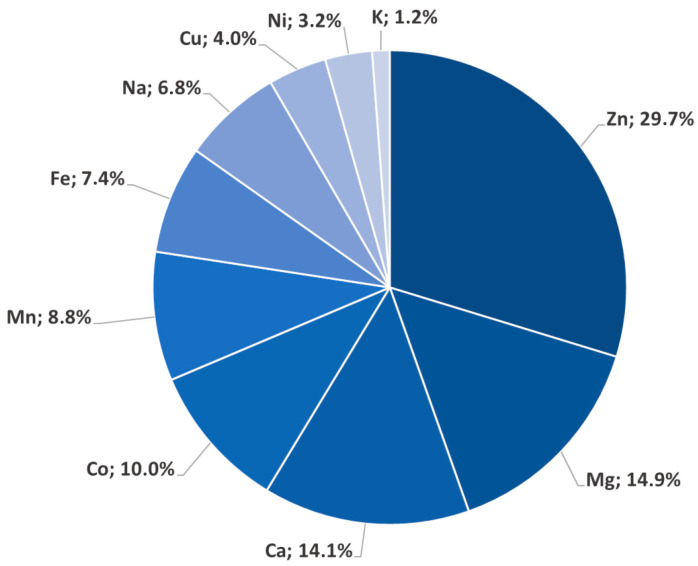
Percentage distribution of metal ions substituted by Cd^2+^, according to the Research Collaboratory for Structural Bioinformatics Protein Data Bank (RCSB PDB).

**Figure 4 biomolecules-13-00036-f004:**
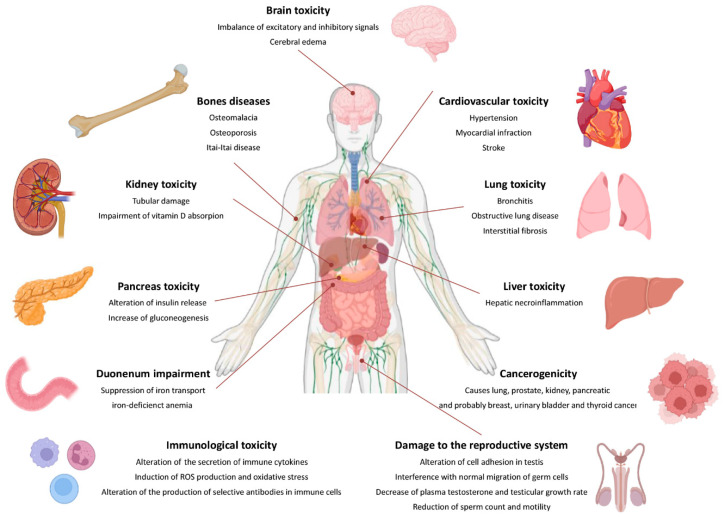
Outcomes of cadmium exposure.

**Figure 5 biomolecules-13-00036-f005:**
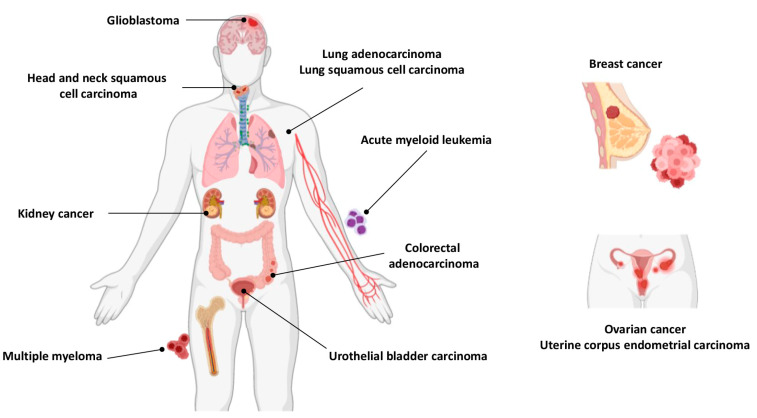
The 12 cancer types associated with specific gene-encoding Cd-metalloproteomes.

**Table 1 biomolecules-13-00036-t001:** Names, UniProt IDs, abundances, and tissue-specific expressions of Cd-binding proteins, annotated as genetic markers in 12 cancer types. Abundances and tissue-specific expressions were mined according to the protein abundance database PaxDB (https://pax-db.org (accessed on 21 October 2022)) and human proteome atlas database (https://www.proteinatlas.org (accessed on 21 October 2022)).

A/A	Gene	UniProt ID	Abundance (ppm)	Protein Name	High Expression Level
1	*GALNT10*	Q86SR1	161	Polypeptide N-acetylgalactosaminyltransferase 10	Lung, gall bladder, kidney
2	*NOTCH4*	Q99466	47	Neurogenic locus notch homolog protein 4	Adipose, lung
3	*GZMB*	P10144	288	Granzyme B	Dendritic cells
4	*BLK*	P51451	36	Tyrosine-protein kinase Blk	Spleen, bone marrow
5	*AXL*	P30530	681	Tyrosine-protein kinase receptor UFO	Testis, skeletal muscle
6	*TGFBR2*	P37173	182	TGF-beta receptor type-2	Adipose, breast
7	*IL2RG*	P31785	105	Cytokine receptor common subunit gamma	Spleen, tonsil
8	*LCK*	P06239	941	Tyrosine-protein kinase Lck	Thymus, T-cells
9	*ESR1*	P03372	35	Estrogen receptor	Endometrium, cervix, uterine
10	*PRKACA*	P17612	697	cAMP-dependent protein kinase catalytic subunit alpha	Cerebral cortex, testis
11	*SHC*	P29353	200	SHC-transforming protein 1	Cerebellum, thyroid gland
12	*HSP90AB1*	P08238	16962	Heat shock protein HSP 90-beta	Cerebellum, adrenal gland

## Data Availability

All data generated or analyzed during this study are included in this article.

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
