# Peer review of "Biological Effects of Human Exposure to Environmental Cadmium"

_biomolecules, 2022, doi:10.3390/biom13010036_

Round 1

Reviewer 1 Report

Cadmium is a metal that is toxic to both human health and the environment and is the subject of much current research. This paper discusses the possible health problems associated with cadmium exposure, in particular its carcinogenicity, and summarizes the treatment and prevention of cadmium poisoning. This review will help readers to gain a more comprehensive understanding of the dangers of cadmium poisoning. However, I think this article needs further revision and here are my suggestions for changes.

1.      The article cites a large number of literature, and it is important to know what has been done, but also to know what are the shortcomings and unsolved problems.

2.      This article includes a lot of content, but so far it seems to be a tedious read. I think you need to focus your article further, as in your article title: Focus on Carcinogenic Effects. Reduce the narrative of other content.

3.      There is too much in the conclusion section, which I think is inappropriate, and you need a summary of the main points of the paper and a brief outline of possible future research directions.

Author Response

  1. We thank the reviewer for his/her comments. As suggested, in the last section “Conclusions” we have summarized the content of the review and what has been done. We also discuss what the unsolved problems are and what still needs to be done to fix them.

  1. We disagree with review 1's statement "This article includes a lot of content, but so far it seems to be a tedious read" as this opinion is purely a subjective perception and not based on the scientific content of the work. We are convinced that the article collects a comprehensive state of the art on the effect and health problems associated of environmental exposure of Cd that will be useful for a wide audience of readers as well as positively commented by the reviewer 2: "This review is well written, and includes wide topics". In consideration of this, we have decided to keep all the topics covered in the text as in the first version, convinced that a further effort to cut some topics would have only weakened the article.

However, we have changed the title to "Biological effects of human exposure to environmental cadmium" so that the reader knows that this is a review that discusses the toxic effects of cadmium in a comprehensive way and not just specifically about its carcinogenic effects, also if it is known to be one of its main features.

  1. The text of the “Conclusion” section has been drastically condensed, as suggested. A brief summary of the main points of the review and an outline of possible future research directions are now presented in a clear manner in the final section.

Reviewer 2 Report

The authors have summarized human exposure to Cd, and they gave introduction about the carcinogenic effects of Cd. This review paper covers description of properties of Cd, environmental dispersion of Cd,  exposure and health effects of Cd, remediation of Cd, and detoxifying reagents. This review is well written, and includes wide topics.

As the title stesses the  carcinogenic effects of Cd exposure, the abstract may need to describe some important facts related with carcinogenic effects of Cd exposure and what are the main reasons for carcinogenic effects.

Author Response

We thank the reviewer for his/her positive comments. As suggested, we improved the abstract including the important facts related with carcinogenic effects of Cd exposure and what are the main reasons for its carcinogenic effects, in this way:

Cadmium (Cd) is a toxic metal for the human organism and for all ecosystems. Cd is naturally found at low levels; however, higher amounts of Cd in the environment result from human activities as it spreads into the air and water in the form of micropollutants as a result of industrial processes, pollution, waste incineration and electronic waste recycling. The human body has a limited ability to respond to Cd exposure, since the metal does not undergo metabolic degradation in less toxic species and is only poorly excreted. The extremely long biological half-life of Cd essentially makes it a cumulative toxin and chronic exposure causes harmful effects from the metal stored in the organs. The present paper considers exposure and potential health concerns on environmental cadmium. Exposure to Cd compounds is primarily associated with an elevated risk of lung, kidney, prostate, and pancreatic cancer. Cd has also been linked with cancers of the breast and urinary bladder. The multiple mechanisms of Cd-induced carcinogenesis include oxidative stress with inhibition of antioxidant enzymes, promotion of lipid peroxidation and interference with DNA repair systems. Cd2+ can replace essential metal ions, including redox-active ones. A total of 12 cancer types associated with specific genes coding for the Cd-metalloproteome were identified in this work. In addition, we summarize the proper treatments of Cd poisoning based on the use of selected Cd detoxifying agents and chelators, and the potential of preventive approaches to counteract its chronic exposure.